# Impact of Government Support on Performing Artists’ Job and Life Satisfaction: Findings from The National Survey in Korea

**DOI:** 10.3390/ijerph17207545

**Published:** 2020-10-16

**Authors:** Hyun-Seung Park, Hyeon-Cheol Kim

**Affiliations:** 1Culture-Specialized Area Development Project Team, Iksan-Cultural & Tourism Foundation, Iksan 54607, Korea; chiwoo978@naver.com; 2School of Business Administration, College of Business and Economics, Chung-Ang University, Seoul 06974, Korea

**Keywords:** performing artist, job satisfaction, life satisfaction, grant

## Abstract

In this study, we aim to propose motives that can help increase the creative activities of Korean performing artists and discuss the policy implications for the sustainable management of Korean performing arts. First, we investigate the characteristics of Korean artists that receive subsidies as a form of government support for undertaking artistic activities. Second, we examine whether receipt of such grants influences the artists’ job and life satisfaction. Through a logistics model, we reconstructed the “2015 Survey Report on Artists & Activities” and validated the research hypothesis. We first considered subsidies that could directly impact artists’ income and activities and then verified whether subsidies influence artists’ job and life satisfaction. As a result of the research, first, art grants should be supported in order to help artists produce creative and experimental works. Second, we showed that artists’ subsidies should be expanded in order to enhance artists’ quality of life and the sustainability of artistic activities. Above all, subsidy support for artists showed that art can be legitimate as a public good, which is a common asset in society.

## 1. Introduction

Organizational structures in the cultural and creative sectors are affected by significant changes in the economic, cultural, and social frameworks in which they operate [1]. Support for artistic/cultural employment and the methods of choosing the (best) artists and cultural organizations to support in an environment of reduced availability of funding are important issues. Regarding the performing arts, many actors must be available for performances and must be hired for long periods. This results in high costs [2,3,4]. The performing arts significantly rely on public partnerships that are not affected by economic crises, as personal support and corporate sponsorships can be heavily impacted by economic crises [1,4]. We seek to assess the multiplicity of problems and challenges that performing artists face in order to suggest this sector’s cultural policies and funding mechanisms. Examining the characteristics of Korean artists that receive grants for their artistic activity can help to determine how the government of Korea is supporting this important sector. This can further allow industry practitioners and government agencies to enact policies that could lead to higher overall creative activity. We propose key elements that motivate artists to increase their creative activity and discuss policy implications for the sustainable management of the performing arts in Korea.

According to Bille et al. [5], artists are known for a profession that presupposes misfortune. According to the results of their research, artists are immensely content with their jobs [5]. However, despite this, artists are known to have high levels of negative emotions such as depression, dissatisfaction, and anger [6]. Recent studies such as that by Botella et al. [7] show that sometimes, artist have negative feelings toward their lives but simultaneously appear to have positive feelings toward their artistic activities [7]. By evaluating their lives, Anand and Kumar [8] emphasize subjective wellbeing, including positive or negative feelings towards specific areas of life and life in general. They also conclude that the level of subjective wellbeing is high as performing arts is a creative profession [8].

From this perspective, it is necessary to discuss sustainability issues in terms of the wellbeing and the quality of life of Korean performing artists. In order to address the issue, we conduct empirical research that will clarify the underlying concepts used to construct a theoretical model and hypotheses. The rest of the paper is organized as follows. First, we introduce the data collection method, sample, and measures used to test the model. We thereafter present the study’s results and discuss its key findings. In conclusion, we provide implications, limitations, and future research directions.

### Background and Hypotheses

Artists and art organizations worldwide, particularly in the field of performing arts, struggle financially [1,3,4,9,10,11,12,13,14]. Unlike in many Western countries, the Korean performing arts are mostly supported by national and public institutions, rather than individuals and companies, which provide employee grants, creative grants for artistic work, practice room grants, etc. [14,15,16,17,18]. The government and public institutions in Korea account for 53.6% of the annual income of performing arts organizations [19].

The “bandwagon effect” is stronger in the field of performing arts than it is in other industries [3]. In this industry, the bandwagon effect refers to a concentration of resources and goods with a small number of superstars [3,20], leaving little for other artists and groups. Compared with groups from other industries with similar levels of education, the average earnings that artists get from their artistic work are low. Hence, they must take on multiple jobs to supplement their earnings [21]. Therefore, as Bille et al. [3] argued, art funds can help to secure stable incomes for artists and act as a motivator for artistic activities, unlike other income streams, such as ticket sales [3].

First, we introduce variables related to the individual characteristics of artists. The resultant analyses aim to determine the role of policies in awarding grants to performing artists. This study attempts to predict the variables that measure the characteristics that increase the likelihood of an artist receiving a grant (public or private). The categorical form of the dependent variables’ logistic regression models is designed for this data. Logistic regression estimates the probability of an event occurring as a function of various explanatory factors.

**Hypothesis 1 (H1).** 
*Which factors among those related to performing artists influence the artists’ likelihood of receiving a grant for their creative activities?*


Second, we investigate the level of job (i.e., arts activities) and life satisfaction from art activities. We examine the feelings of wellbeing and happiness that can motivate performing artists to continue their careers. Particularly, we study whether they are satisfied with their jobs and lives depending on whether they receive public or private subsidies. In conclusion, the influence relationship is investigated through regression analysis in which there is a difference in the degree of satisfaction depending on the performing arts genre.

**Hypothesis 2 (H2).** 
*Does receiving subsidies (including both public and corporate) significantly influence the lives of performing artists and their satisfaction with artistic activities?*


**Hypothesis 3 (H3).** 
*According to the performing arts genre, do artists’ degrees of satisfaction with their jobs and lives significantly differ?*


## 2. Materials and Methods

The data used to test the hypotheses are from the “2015 Survey Report on Artists & Activities,” that the Republic of Korea’s Ministry of Culture, Sports and Tourism [8] has conducted among cultural artists every three years since 1988. However, the amendment made to the Act on the Welfare of Artists in 2013 established new legal grounds for the surveys on the status of artists. Since then, the survey subjects have been amended to 131,332 cultural artists engaged in 14 cultural arts fields, including literature, art, architecture, photography, music, popular music, Korean music, dance, drama, and film. A stratified random sampling method was used to choose 5008 participants from each of the 14 fields and 16 Korean provinces. The survey inquired about the conditions of artistic activity, satisfaction with the field, and demographic characteristics. Therefore, this survey of artists was the first to be carried out with the full reorganization of its name, size, and method in the Republic of Korea.

This study analyzes five areas of the performing arts among the 14 cultural art fields: music, popular music, traditional music, dance, and theater. Among the 1979 samples, the observations that were said to have responded to the questionnaire unfairly were excluded, leaving 1920 observations for the analysis. Equations (1)–(3) presents the basic logistic model used in the estimation for receiving grants (public and private):(1)$$\log\frac{P(y=1|x_{1},\;x_{2},\;x_{3},\;\cdots ,\;x_{i})}{1-P(y=1|x_{1},\;x_{2},\;x_{3},\;\cdots ,\;x_{i})}=\alpha +\beta_{1}x_{1}+\beta_{2}x_{2}+\cdots +\beta_{i}x_{i}+\varepsilon$$
(2)$$\mathrm{where}\;\log(\frac{P_{i}}{1-P_{i}})\;\mathrm{is}\;Logit\left(P_{i}\right)$$
(3)$$P_{i}=\frac{1}{1+e^{-\left(\beta_{0}+\beta_{1}X_{1}+\beta_{2}X_{2}+\cdots +\beta_{i}X_{i}\right)}}$$

Table 1 describes the two sets of independent variables used to estimate the model. The dependent variable was receipt of support for creative grants over the past year: 1 = received at least one grant (including public and private) and 0 = received no grants.

Table 2 shows the items used for both job and life satisfaction, which were used as dependent variables. These questions were measured on a 5-point Likert scale, with two questions for life satisfaction and two for artistic activity.

## 3. Results

The data used in this study surveyed artists from 14 art genres, including Korean literature, art, architecture, photography, music, popular music, Korean music, dance, drama, and film. This study is the most labor-intensive study of these 14 cultural arts fields. This study is limited to the performing arts field, which is heavily influenced by the socio-economic environment, but the results are important because it examines the role of subsidies in this field. 

The testing of Hypothesis 1 reveals that performing artists’ socio-economic variables, such as gender, age, education level, and full-time occupational status, have a decisive effect on the receipt of grants. Table 3 contains the results of the basic logistic regression model, which reports the effect of the independent variables.

An auxiliary regression view predicted values as explanatory variables and the actual outcome as a response variable. Hat and hat squared are coefficients of the predicted values and quadratic term, respectively. The model correctly specified 79.2% of the total cases. The chi-square value examines the model fit of logistic regression, which is further examined through Hosmer and Lemeshow tests. The chi-square of the model was 148.94 at a 0.001 significance level. This indicates that the model was overall significant. Seven of the twelve explanatory variables included in the logistic regression model significantly predicted the recipients of grants. Table 3 shows that, while JOB(full or part time artists), OWN(have a practice room), EDU(education level), AGE(age level), COST(learning and training cost), SPEND(spending on artistic works), and WORK(yearly number of artistic works) are positively associated with receiving grants, GENDER(male or female), CAREER(career interruption experience), COPY(have copyright or not), MEMBER(join an association or union), and DEBUT(activity period after debut) are not associated with the respective reception of grants. JOB and OWN were statistically significant at the 5% significance level, while full-time artists were likely to receive higher grants than part-time artists. The received grant increased by 27.4% if the odds ratio was for full-time artists. This result means that full-time artists are 27.4% more likely to receive grants than are part-time artists. Artists that own practice rooms received more grants than those that did not-own practice rooms. The odds ratio was increased by 39.2%.

Among the demographic variables, more educated and older artists were most likely to receive grants. With the EDU 3 variable (association degree), the odds of receiving a grant increased by 116.4%. The older the artist, the greater the odds of receiving a grant. Additionally, as for AGE, the odds ratio was statistically high in all the variables. This indicated that the older the artist, the higher the probability of receiving subsidies.

In conclusion, the artistic variables COST and WORK were statistically significant at 10%, while SPEND was within 1%. Overall, learning and training costs, spending on creating artistic works, and considerable artistic activity increased the probability of receiving a grant.

Table 4 shows the results of the regression analysis of both artists’ job and life satisfaction, according to whether grants are received for each performing arts genre, to test Hypotheses 2 and 3.

As a result of the analysis, it is clear that artists that received grant funds had a high degree of job satisfaction (t = 5.461, *p* < 0.01), and that their satisfaction with their life satisfaction was statistically significant (t = 1.695, *p* < 0.1). Compared to artists of other genres, the life satisfaction of Genre 4 (dance) artists’ lives was statistically significantly higher (t = 1.996, *p* < 0.05). The life satisfaction of artists in other genres was not statistically significant. Further, job satisfaction did not show significant satisfaction in all genres.

## 4. Conclusions and Implications

Most empirical studies of performing artists in Korea are descriptive and elaborately explain how different socio-economic characteristics relate to income. However, these studies did not first consider grants that could directly impact the artist’s income and activities. Second, research on whether subsidies affect the job and life satisfaction of artists has been lacking.

First, this study shows that the likelihood of an artist receiving grants is more important than other personal traits in terms of learning and training costs, spending on works of art, and the number of works of art produced. In other words, it is clear that the substantial benefits of grants for artists with large amounts of production and investment in creating works of art and learning and training for performances means that the artists are more active in receiving grants to make their own works. The importance of the cost and expenditure of learning and training for a work of art is amplified by the amount of art created. The levels of performing arts may fluctuate as participating in one form of art increases the likelihood of participation in another. These results support the view that the economic resources gained through subsidies are currently a major factor in determining inequality in the production of art. Therefore, policies aimed at increasing the number of works of art and securing the stability of performing artists should consider the artistic characteristics of artists specialized in art. Therefore, to ensure creative and experimental performances, art grants must be provided in the full range from covering learning and training expenses to spending on artwork intended to support labs. The number of works of art is partly determined by complementary effects. Therefore, the more an artist engages in one form of performing art, the more likely they are to engage in other forms of performing art. AGE was the only variable among the demographic variables that played a decisive role in the likelihood of an artist receiving grants. In particular, among the demographic variables, the DEBUT variable did not secure statistical significance. It is significant that the variable age, not the debut year, plays the role of subsidy beneficiaries, and this fact seems to require a multilateral analysis of the equity of subsidy beneficiaries. In particular, age and artistic resources need to be considered in more detail. For example, it is necessary to look at the results of having a practice room as a resource that plays a positive role in receiving grants. This may mean that the higher the amount of socioeconomic capital or resources possessed, the higher the probability of receiving subsidies, rather than the original positive role in the activities of artists. This suggests that studies that specifically examine negative views or differentiation of subsidies in the future are also necessary. Specifically, to fully understand what support is needed and when it is most effective requires funding and other forms of support at various stages of artists’ practical careers, rather than the artist’s age and the amount of social capital, such as a practice room, held by the artist. This is because the age of the artist and the amount of social capital to be supported can act as inequalities in receiving grants.

Second, the high satisfaction with job and life amongst artists that received grants is the subjective level of welfare that they can feel through various forms of support that help to increase their levels of satisfaction with their artistic activities and lives. That is, the aspect of expanding the artists’ wellbeing and quality of life and, through this, the performing arts. This means that private and public subsidies must be expanded for sustainability. It also shows that public and private subsidies should be expanded as a sustainability issue in terms of the wellbeing and quality of life of Korean performing artists. What role the state should play and to what extent to support the arts field differs depending on history, value, the economic level of each country, and cultural support policies. Most countries are striving to promote, protect, and nurture art through various support policies and to either directly lead the creation of art or indirectly support it through financial aid [22,23]. Art can be viewed as an important common social asset that develops the economy and sustains society as a source of creative technological innovation [24,25,26,27,28,29]. Furthermore, from an infrastructure perspective, government support for this sector can be justified in terms of the supply of public goods.

To sum up this study’s results, the arts field is exposed to employment insecurity and poor creative environments due to high labor flexibility, intermittent employment opportunities, and the small art market [30,31]. Therefore, subsidies are essential for artists’ continued artistic activities. In particular, the result that receiving art subsidies positively impacts the job and life satisfaction of artists’ can be said to indicate the importance of subjective welfare. This also improves the wellbeing of artists and of a creative environment of artists in the future.

## Figures and Tables

**Table 1 ijerph-17-07545-t001:** List of independent variables.

Variable	Description
Gender	Gender: Dichotomous variable, 0 = Female, 1 = Male
Job	Full-Time Artist: Dichotomous variable, 0 = Part-time, 1 = Full-time
Career	Career Interruption: Dichotomous variable, 0 = No, 1 = Yes
Copy	Copyright Holding: Dichotomous variable, 0 = No, 1 = Yes
Member	Membership in Association or Union: Dichotomous variable, 0 = No, 1 = Yes
Own	Have a Practice Room: Dichotomous variable, 0 = No, 1 = Yes
Edu	Education Level: Categorical variable dichotomized in final analysis, 1 = Less than high school, 2 = In college or university, 3 = Associate degree or higher
Age	Age: Categorical variable dichotomized in final analysis, 1 = 20–29, 2 = 30–39, 3 = 40–49, 4 = 50–59, 5 = 60+
Debut	Activity Period After Debut: Categorical variable dichotomized in final analysis, 1 = 0–4 years, 2 = 5–10 years, 3 = 11–20 years, 4 = 21–30 years, 5 = 30+ years
Cost	Learning and Training Cost: Categorical variable dichotomized in final analysis, 1 = None, 2 = $1–$940, 2 = $941–$4700, 3 = $4701+
Spend	Spending on Artistic Works: Categorical variable dichotomized in final analysis, 1 = None, 2 = $1–$940, 2 = $941–$4700, 3 = $4701+
Work	Yearly Number of Artistic Works: Categorical variable dichotomized in final analysis, 1 = None, 2 = 1–5 works, 2 = 6–10 works, 3 = more than 11 works
Genre	Korean Performing Arts Genre: Categorical variable dichotomized in final analysis, 1 = pop music, 2 = music, 2 = Korean traditional music, 3 = dance, 4 = theater

**Table 2 ijerph-17-07545-t002:** List of dependent variables.

Variables	Survey Items
Job (i.e., arts activities) satisfaction	I’m satisfied with my artistic activities.
	I think my work is very valuable.
Life satisfaction	I’m generally satisfied with my recent life.
	I feel very happy these days.

**Table 3 ijerph-17-07545-t003:** Results of logistic regression analysis.

Variable	B	SE	Wald	df	*p*	Exp (B)
Gender	0.143	0.123	1.347	1	0.246	1.153
Job	0.242	0.121	4.032	1	0.045	1.274
Career	0.080	0.164	0.238	1	0.625	1.083
Copy	0.047	0.156	0.089	1	0.765	1.048
Member	–0.022	0.167	0.017	1	0.895	0.978
Own	0.331	0.123	7.235	1	0.007	1.392
Edu1						
Edu2	–0.187	0.180	1.088	1	0.297	1.206
Edu3	–0.772	0.200	14.856	1	0.000	2.164
Age1						
Age2	0.502	0.217	5.374	1	0.020	1.653
Age3	–0.806	0.242	11.051	1	0.001	2.238
Age4	1.010	0.252	16.022	1	0.000	2.745
Age5	–0.770	0.287	7.218	1	0.007	2.159
Debut1						
Debut2	–0.112	0.215	0.272	1	0.602	1.119
Debut3	–0.012	0.212	0.003	1	0.956	0.988
Debut4	–0.245	0.247	0.985	1	0.321	0.783
Debut5	0.030	0.268	0.013	1	0.909	1.031
Cost1						
Cost2	0.159	0.093	2.931	1	0.087	1.172
Cost3	0.031	0.053	0.351	1	0.554	1.032
Cost4	0.090	0.050	3.239	1	0.072	1.094
Spend1						
Spend2	0.059	0.100	0.342	1	0.559	1.061
Spend3	0.165	0.052	10.147	1	0.001	1.179
Spend4	0.206	0.044	22.205	1	0.000	1.229
Work1						
Work2	0.980	0.566	2.999	1	0.083	2.665
Work3	0.593	0.546	1.178	1	0.278	1.810
Work4	1.135	0.595	3.632	1	0.057	3.111
Con	–3.633	0.612	35.227	1	0.000	0.026
Model Fit Statistics		Likelihood ratio χ^2^ = 148.94, df = 25, *p* < 0.001, *n* = 1,920, −2 Log Likelihood = −1826.77, Nagelkerke R^2^ = 0.12, correctly classified = 79.2%				

Note. S.E., standard error; Exp (B), multinomial logistic regression.

**Table 4 ijerph-17-07545-t004:** Results of regression analysis.

Variable	Job Satisfaction				Life Satisfaction			
	B	SD	*β*	t	B	SD	*β*	t
Grant Received	0.214	0.039	0.124	5.461 ***	0.073	0.043	0.039	1.695 *
Genre1								
Genre2	0.040	0.052	0.024	0.772	−0.38	0.057	−0.21	−0.668
Genre3	0.080	0.052	0.047	1.539	0.88	0.057	0.047	1.541
Genre4	0.003	0.056	0.002	0.055	0.121	0.061	0.059	1.996 **
Genre5	0.072	0.051	0.044	1.414	−0.03	0.055	−0.002	−0.054
Con	3.662	0.039		92.75 ***	3.462	0.043		80.138 ***
R^2^	0.018				0.007			
adjR^2^	0.160				0.005			
*F*	7.062 ***				2.861 **			

Note. *, *p* < 0.1; **, *p* < 0.05; ***, *p* < 0.001; S.D., standard deviation.

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
