# Peer review of "Impact of Government Support on Performing Artists’ Job and Life Satisfaction: Findings from The National Survey in Korea"

_ijerph, 2020, doi:10.3390/ijerph17207545_

Round 1

Reviewer 1 Report

Dear Editors of IJERPH,

Thank you for inviting me to review this manuscript by Hyun-Seung Park and Hyeon-Cheol Kim.

This manuscript makes an original contribution and needed knowledge for understanding the variables associated with Korean government grants received by Korean artists and life satisfaction as an artist.

I will not comment on the methodology as I am not trained in the methodology that the authors use. This is also why I completed "Not applicable" for research methodology and design. I focused on the framing of the study, including scholarly evidence for the framing, the results and conclusion.

My specific comments are below and I have given the corresponding Line of the manuscript for reference.

Line 59: More citations are needed to provide evidence for the claim that “¸Artists and art organizations worldwide, particularly in the field of performing arts, struggle with financial difficulties.” “Worldwide” requires evidence. Some recommended sources to be considered/cited are below. Further research and citations are needed.

Jung, Kwangho, and M. Jae Moon. "The double‐edged sword of public‐resource dependence: The impact of public resources on autonomy and legitimacy in Korean cultural nonprofit organizations." Policy studies journal 35.2 (2007): 205-226.

Lee, Hye-Kyung. "Progress without consensus:‘instituting’Arts Council in Korea." International journal of cultural policy 18.3 (2012): 323-339.

McAndrew, Clare, and Cathie McKimm. "The living and working conditions of artists in the Republic of Ireland and Northern Ireland." The Arts Council/An Chomhairle Ealaíon and Arts Council of Northern Ireland (2010).

Sandals, Leah. “Artists and Advocates Push for Universal Basic Income in Canada,” Canadian Art, July 2020, accessed September 23, 2020, https://canadianart.ca/news/artists-and-advocates-push-for-universal-basic-income-in-canada/.

Shin, Hyesun, and InSul Kim. "Are Public-Led Arts Incubating Programs a Double-Edged Sword? A Case Study of the Arts Council Korea’s Performing Arts Grant Program." The Journal of Arts Management, Law, and Society 49.1 (2019): 89-103.

Solhjell, Dag. "Poor artists in a welfare state: a study in the politics and economics of symbolic rewards." International Journal of Cultural Policy 7.2 (2000): 319-354.

Throsby, David. "Preferred work patterns of creative artists." Journal of Economics and Finance 31.3 (2007): 395-402.

Line 131: The statement of significance requires prior knowledge of the meaning of the values. Citation needed.

Line 133-134: It’s not clear what the all the all-capital words refer to (i.e. OWN, COST, SPEND, DEBUT, MEMBER, etc.). The meaning of these variables need to be provided. Including these capitalized codes in Table 1 would be helpful (i.e. replacing the Variables listed with the all-capped variables in the text, or making a decision for consistency and ease of reading).

Line 138: The sentence “The received grant increases by…full-time artists” needs to be clearer.

Lines 124-127: This reader wanted to know the results based on genre and whether there were statistically significant differences between genres. If genres are not discussed separately or given some attention, the reader asks why the authors do not report according to the description of the study in Lines 103-104 (“five areas of the performing arts among the 14 cultural art fields: music, 103 popular music, traditional music, dance, and theater”).

Line 163: The context needs to be clear, i.e. Republic of Korea.

Line 167-168: This sentence does not make sense given that the manuscript states “Overall, learning and training costs, spending on creating artistic works, and considerable artistic activity increased the probability of receiving a grant,” In Line 146-147.

Regarding the analysis in Lines 167-180, the authors need to consider the career point of an artist, e.g.. emerging, mid-career, or senior artist in their conclusions. Perhaps, these categories need to be considered in the government grants. Because artists with resources are more successful in grants, the question arises whether awards are made on the basis of where an artist is at in their career. The article should state the award categories given nationally and do some corroborative analysis with their findings, or point to this area as future research. Otherwise, the authors’ findings may not have limited policy relevance or influence.

Line 181: The sentence needs editing. What is meant by “they need subjective welfare for the artist?” This is unclear.

Line 182-183: same as above in Line 181. This sentence needs editing for clarity.

Line 186: There is no evidence provided in the paper for this claim. Citations are needed.

Line 188-180: The authors write: "Art can be viewed as an important social common asset that develops the economy and sustains society as a source of creative technological innovation." Same as above. This is a broad claim and needs to be substantiated by scholarly evidence. Citations are needed.

Line 191-192: The authors write: "the arts field is exposed to employment insecurity and poor creative environments due to high labor flexibility, intermittent employment opportunities, and the small art market." This is a broad claim and needs to be substantiated by scholarly evidence. Important considerations for poor creative environments are environmental barriers, continuous cuts to arts funding since the 1990s and the positioning of artists by governments and arts councils increasingly as economic entrepreneurs or economically oriented (Changfoot, 2007) instead of positioning artists as exemplars of cultural expression as well as social and political goals.  Citations are needed. Include:

Bang, Gui Hee, and Kyung Mee Kim. "Korean disabled artists’ experiences of creativity and the environmental barriers they face." Disability & Society 30.4 (2015): 543-555.

Changfoot, Nadine. "Local activism and neoliberalism: performing neoliberal citizenship as resistance." Studies in political economy 80.1 (2007): 129-149. See pp. 133-134.

Throsby, David, and Anita Zednik. Do you really expect to get paid?: an economic study of professional artists in Australia. Australia Council for the Arts, 2010.

Line 196: delete rogue “s”

I enjoyed reading this manuscript and I hope these comments are helpful to the authors. Again, thank you for inviting me to read this manuscript. Knowledge of the variables influencing the award of grants, artist life satisfaction and artistic satisfaction is very important within each nation and across nations.

Sincerely,
Reviewer

Author Response

Please find attached the point-by-point responses to your comments. Thank you for this encouragement and all your comments.

Reviewer 2 Report

The paper is clean and easy to follow. No suggestions, except that I would add punctuation (commas or periods) after the mathematical equations to correctly incorporate them into the sentences  to which they belong.

Author Response

(The authors gave the same response as above.)

Reviewer 3 Report

Thank you for giving me the opportunity to review the article „Impacts of government support on performing artists’ job and life satisfaction: Findings from the national survey in Korea”. Below my remarks:

  • “The purpose of this study is to propose motives to increase the creative activities of Korean performing artists and discuss policy implications for the sustainable management of Korean performing arts”. However, I think that the main aim of the work should be more scientific. It needs to be rewritten;
  • The Authors should carry out a solid review of the literature and present the results obtained. The current references are very poor;
  • The statistical analyses carried out are very poor. They should be deepened;
  • The discussion and conclusions are not scientific in nature. They should be compared with the results obtained by other researchers;
  • In my opinion, the article should be completely rewritten. The Authors should definitely indicate the scientific context. Currently, the article is a typical simple research report.

Author Response

(The authors gave the same response as above.)

Round 2

Reviewer 1 Report

The manuscript has been significantly improved and now warrants publication in IJERPH with three minor revisions:

First:

190 ...The fact that the variable age rather than the year of debut plays

191 a role in subsidy beneficiaries is signifcant seems to be a new look. Particularly, age and artistic resources needs to be considered in more detail. For example, it is necessary to look at the results of

192 having a practice room as a resource that plays a positive role in receiving grants.

Second:

For the lines below, the authors need to explain or define “the negative views or differentiation of subsidies” and why it’s necessary to examine these.

194 This suggests that studies that

195 specifically examine the negative views or differentiation of subsidies in the future are also necessary.

Third: the page numbers to the article by Changfoot N. are 129-148 (not 133-134).

Author Response

Please find attached the point-by-point responses to the reviewers’ comments.

Reviewer 3 Report

I thank the Authors for including my comments in the article.

Author Response

Thank you very much for your attention and support to our research.